

**Seasonal variations in composition and sources of atmospheric ultrafine particles**
**in urban Beijing based on near-continuous measurements**
*Xiaoxiao Li[1,2], Yijing Chen[1], Yuyang Li[1], Runlong Cai[3], Yiran Li[1], Chenjuan Deng[1], Chao Yan[3,4],*
*Hairong Cheng[2], Yongchun Liu[4], Markku Kulmala[3,4], Jiming Hao[1], James N. Smith[5*], and Jingkun*
*Jiang[1*]*
[1] State Key Joint Laboratory of Environment Simulation and Pollution Control, School of Environment,
Tsinghua University, 100084 Beijing, China
[2] School of Resources and Environmental Sciences, Wuhan University, 430072 Wuhan, China
[3] Institute for Atmospheric and Earth System Research / Physics, Faculty of Science, University of
Helsinki, 00014 Helsinki, Finland
[4] Aerosol and Haze Laboratory, Beijing Advanced Innovation Center for Soft Matter Science and
Engineering, Beijing University of Chemical Technology, 100029 Beijing, China
[5] Chemistry Department, University of California, Irvine, CA 92697, USA
*Correspondence to:* Jingkun Jiang (jiangjk@tsinghua.edu.cn) and James N. Smith
(jimsmith@uci.edu)
**Abstract.** Understanding the composition and sources of atmospheric ultrafine particles (UFPs) is
essential in evaluating their exposure risks. It requires long-term measurements with high time
resolution, which are to date scarce. We performed near-continuous measurements of UFP composition
during four seasons in urban Beijing using a thermal desorption chemical ionization mass spectrometer,
accompanied by real-time size distribution measurements. We found that UFPs in urban Beijing are
dominated by organic components, varying seasonally from 68 to 81%. CHO organics are the most
abundant in summer, while sulfur-containing organics, some nitrogen-containing organics, nitrate, and
chloride are the most abundant in winter. With the increase of particle diameter, the contribution of
CHO organics decreases, while that of sulfur-containing and nitrogen-containing organics, nitrate, and
chloride increase. Source apportionment analysis of the UFP organics indicates contributions from
cooking and vehicle sources, photooxidation sources enriched in CHO organics, and
aqueous/heterogeneous sources enriched in nitrogen- and sulfur-containing organics. The increased
contributions of cooking, vehicle, and photooxidation components are usually accompanied by
simultaneous increases in UFP number concentrations related to cooking emission, vehicle emission,
and new particle formation, respectively. While the increased contribution of the
aqueous/heterogeneous composition is usually accompanied by the growth of UFP mode diameters.
The highest UFP number concentrations in winter are due to the strongest new particle formation, the
strongest local primary particle number emissions, and the slowest condensational growth of UFPs to
larger sizes. This study provides a comprehensive understanding of urban UFP composition and
sources and offers valuable datasets for the evaluation of UFP exposure risks.



## 1. Introduction

Ultrafine particles (UFPs, particles with diameters smaller than or equal to 100 nm) have significant effects on human health (HEI, 2013; Ohlwein et al., 2019; WHO, 2013) and global climate (Kulmala et al., 2004; Pierce and Adams, 2007). Their human exposure risks and climate effects are highly related to their composition and size (Oberdorster et al., 2005; Pierce and Adams, 2007). To better evaluate the exposure risks of UFPs and to formulate corresponding air quality guidelines, the World Health Organization made several recommendations to guide measurements and regulations of UFPs in 2021 (Organization, 2021). They emphasized that local understanding of UFP origins and their chemical composition are scarce in most parts of the world.

Current field studies of atmospheric UFP composition and their source apportionment are mostly based on offline sampling. These measurements usually use a size-resolving impactor to collect UFPs on filters for tens of hours to several days (Cabada et al., 2004; Cass et al., 2000; Ham and Kleeman, 2011; Herner et al., 2005; Kleeman et al., 2009; Massling et al., 2009; Xue et al., 2019; Xue et al., 2020a; Xue et al., 2020b; Zhao et al., 2017). They found that organic carbon, sulfate, and nitrate could account for 50-90% of the detected compounds, and the composition could vary greatly with UFP sizes due to different sources and atmospheric evolutions (Cabada et al., 2004; Herner et al., 2005; Massling et al., 2009). For source apportionment, the Chemical Mass Balance analysis was often used which relies heavily on the source profiles and the limited numbers of molecular markers (Ham and Kleeman, 2011; Kleeman et al., 2009; Xue et al., 2019). The identified sources include meat cooking, gasoline, diesel, motor oil, and wood burning. However, these offline explorations can not capture the high temporal variability in size-resolved UFP composition and sources, nor can they distinguish primary UFPs from secondary sources, e.g., new particle formation, due to the lack of tracers and composition profiles for secondary sources. Simultaneous real-time measurements of UFP composition and size distributions are needed to accurately identify both primary and secondary sources.

Using a thermal desorption chemical ionization mass spectrometer (TDCIMS), the size-resolved composition of UFPs can be measured with a resolution of tens of minutes (Li et al., 2021; Smith et al., 2004). Previously, it has been used in a number of sites for short-term measurements and found distinct characteristics for UFPs in urban (Li et al., 2022a; Li et al., 2021; Smith et al., 2008; Smith et al., 2005), rural (Lawler et al., 2020; Smith et al., 2010), and remote areas (Glicker et al., 2019; Lawler et al., 2018; Lawler et al., 2021; Lawler et al., 2014). For instance, UFPs in urban areas have more nitrogen- and sulfur-containing organics (Smith et al., 2005; Winkler et al., 2012), while those at forest sites have more CHO organics (Glicker et al., 2019; Lawler et al., 2018). With these near-continuous measurements, unique sources such as fungal bursts (Lawler et al., 2020) and sea-salt nanoparticles (Lawler et al., 2014) were identified and the mechanisms of new particle formation were examined (Li et al., 2022a). Source apportionment analysis was performed for the high time-resolution TDCIMS results in Amazon Basin to isolate anthropogenic UFPs from background UFPs (Glicker et al., 2019).



These analyses mainly focused on short-term analysis covering several weeks. To address the UFP
composition and sources from a more comprehensive view, there is an urgent need for long-term and
high-time-resolution measurements in diverse environments.
The primary and secondary sources of particles in urban atmospheres usually show significant seasonal
characteristics. Thus, addressing the seasonal variations of UFPs, as well as their governing factors, is
fundamental to evaluating their long-term impacts. For fine particles in Beijing, coal combustion is
more abundant in winter due to domestic heating in the surrounding regions (Sun et al., 2015; Zhang
et al., 2013), biomass burning is more abundant in harvest seasons (Zhang et al., 2008), and dust storms
are more frequent in spring (Xu et al., 2020; Zhang et al., 2013). Besides these primary sources,
previous studies on larger particles showed higher oxidation states of organic aerosols in summer due
to stronger photochemical processes (Hu et al., 2017; Ma et al., 2022; Sun et al., 2018). However,
considering the short lifetime of UFPs, seasonal variations of the composition and sources of UFPs are
likely different. For example, as an important source of UFPs, new particle formations in Beijing were
observed to be the weakest in summer and strongest in winter due to temperature variations (Deng et
al., 2020; Li et al., 2020; Wu et al., 2007). Similar to Beijing, seasonal variations of UFP composition
based on high time-resolution measurements are also scarce for other atmospheric sites.
Here, we performed near-continuous measurements of UFP composition and size distributions over
four seasons in a typical megacity of Beijing with ~22 million people. The UFP composition, its size
dependence, and seasonal variability were analyzed. Several molecular markers from cooking and
vehicle emissions were identified. These markers were combined with the Positive Matrix
Factorization (PMF) analysis to address contributions from primary and secondary sources of UFPs.
The aerosol General Dynamic Equation (GDE) was used to quantify the emission rates of primary
UFPs and the formation rates of secondary UFPs. The driving factors for the seasonal variations of
UFP composition and number concentrations were identified.
**2.  Methods**
**2.1 Field measurements.**
The sampling site is on the fifth floor of a building on the west campus of Beijing University of
Chemical Technology (39°94′N, 116°30′E) (Liu et al., 2020). The site is a typical urban site,
surrounded by residential and commercial buildings. Three trafficked roads are 130~565 m away from
the sites. UFP composition, particle number-size distribution, trace gases, and meteorological
conditions were measured over four seasons between Dec. 2019 and Aug. 2021. An overall of 149
days' TDCIMS measurements were used for analysis, with at least three weeks' data for each season.
Details of the sampling periods are described in Table S1.
UFP composition was measured by the TDCIMS using the "bulk collection mode" (Li et al., 2021;
Smith et al., 2004). The TDCIMS collects pre-charged particles on a high voltage-biased Pt filament



and then vaporizes the particles for analysis by a chemical ionization high-resolution time-of-flight
mass spectrometer (CI-HTOF, Aerodyne Research Inc. and Tofwerk AG). The particle electrostatic
collection efficiency on the filament decreases rapidly with increasing particle size, ensuring that the
collected particle mass is mainly from UFPs (Li et al., 2021). Using $O_2^-$ as the reagent ion, sulfate,
nitrate, chloride, and most of the oxygenated organics can be measured, while black carbon,
hydrocarbon compounds, and bases such as ammonium and aminium cannot be detected. Every
sampling cycle is followed by a background cycle where no voltages are applied to the Pt filament for
particle collection. The signals from the background cycle are subtracted from the sampling cycle to
exclude minor influences from the gas phase compounds. Each analysis cycle (including a sample and
a background cycle) is set to be 10-40 min, depending on the estimated sample mass. The detailed
principles, operations, and quantifications of the TDCIMS are the same as the "bulk collection mode"
described in our previous study (Li et al., 2021).
The particle number size distributions from 1 nm to 10 μm were measured using a home-built particle
size distribution system (PSD, 3 nm–10 μm) and a diethylene glycol scanning mobility particle
spectrometer (DEG-SMPS, 1–7.5 nm). The configuration and operation of the PSD are the same as
described in our previous studies (Cai et al., 2017). The time resolution of the measurement is 5 min.
The number and mass concentrations of atmospheric UFPs were estimated via the integration of size
distribution measurements, assuming spherical particles with a density of 1.4 g cm$^{-3}$.
Other parameters used in this study include the meteorological conditions measured by the
meteorology stations (AWS310, Vaisala Inc., Finland) and trace gases measured by the trace gas
analyzers (TGA, Thermo Fisher). The mixing layer height (MLH) was estimated from the vertical
profiles measured by a ceilometer (CL51, Vaisala Inc., Finland) and a three-step idealized-profile
method was used to estimate the MLH (Eresmaa et al., 2012).

**2.2 Source apportionment of UFP composition.**

The Igor-based interface SoFi (solution finder, version 6.5) and ME-2 (Canonaco et al., 2013) were
used for the PMF analysis to analyze the sources of organics in UFPs. The integrated thermal
desorption signals of organic peaks with m/z between 100 and 300 measured by the TDCIMS were
used as the input data matrix. The integrated thermal desorption signals from the background samples
were used to derive the input error matrix. The best solution in each season was chosen according to
$Q/Q_{exp}$, the similarities between m/z profiles, time series, and diurnal variations of the factors. The
correlations between each factor and the measured key species, trace gas, and PM$_{2.5}$ were calculated
for better identification of the factors. It should be noted that there were also many peaks with m/z
below 100, but a large fraction of them was from thermal decomposition, and their inclusion would
add great complexity to the factor assignments. The signal intensity instead of the mass concentration
was used because sensitivity quantification of the TDCIMS was based on the calibration of limited
numbers of compounds which may induce unknown uncertainties when quantifying the sources. As a



result, the signal intensity measured by the TDCIMS is reported for reference and the relative
variations of detected species are studied rather than their estimated ambient concentrations.
**2.3 Quantifying source and loss terms of UFP number concentrations.**
In the measured size distribution plots, there are usually abrupt increases in UFP number concentration.
During new particle formation (NPF) periods, the abrupt increases of UFPs are usually accompanied
by a burst of sub-3 nm particles and usually start from noontime. During non-NPF periods, the abrupt
increases of UFPs are usually accompanied by an increase in primary emission tracers (as will be
shown in Section 3.2). We apply the GDE to quantify the new particle formation rates ($J$) and primary
particle number emission rates ($E$) at the observation site. The calculation of $J$ follows those described
in previous studies (Cai and Jiang, 2017; Cai et al., 2017). The calculation of $E_{[i,j]}$ (m$^{-3}$ s$^{-1}$), the particle
emission rates in the size range of [$d_i$, $d_j$], follows Eq. 1 (Cai et al., 2018; Kontkanen et al., 2020)
during non-NPF periods.
$$E_{[i,j]} = \frac{dN_{[i,j]}}{dt} + GR(n_j\text{-}n_i) + CoagSnk_{[i,j]} - CoagSrc_{[i,j]} - TR_{[i,j]} \qquad \text{(Eq. 1)}$$

Where  $\dfrac{dN_{[i,j]}}{dt}$  (m$^{-3}$ s$^{-1}$) is the variation of the particle number concentration in the size range of [$d_i$,
$d_j$] during the period of d$t$ (s$^{-1}$); $GR(n_j\text{-}n_i)$ (m$^{-3}$ s$^{-1}$) is the net condensation growth term, $GR$ (m s$^{-1}$) is
the condensational growth rate of particle $d_i$, and $n_i$ (m$^{-4}$) is the particle number size distribution
function for particle $d_i$; $CoagSrc_{[i,j]}$ and $CoagSnk_{[i,j]}$ (m$^{-3}$ s$^{-1}$) are the coagulation source and sink terms;
$TR_{[i,j]}$ is the transport term. Overall, 33 size bins were included in the size range of 3-50 nm.
The term  $\dfrac{dN_{[i,j]}}{dt}$  and $CoagSnk_{[i,j]}$ can be directly calculated from the size distribution data (Cai et al.,
2018). $GR$ is calculated by the theoretical condensation of the condensable vapors, that is the sum of
H$_2$SO$_4$ and condensable organic vapor concentrations. Here, we regard condensable organic vapors as
oxygenated organic molecules (OOMs) with saturation vapor pressure lower than 0.3 μg m$^{-3}$ as in our
previous studies (Li et al., 2022a; Qiao et al., 2021). Since not all the observation days were equipped
with the measurements of condensable vapors, we adopted seasonal-dependent $GR$ derived from
seasonal average condensable vapor concentrations reported in our previous study, that is $1.2\times10^7$,
$9.9\times10^7$, $1.2\times10^8$, and $5.0\times10^7$ cm$^{-3}$ for winter, spring, summer, and autumn (Qiao et al., 2021),
respectively. For particles smaller than 50 nm, $CoagSrc_{[i,j]}$ term can be neglected; for particles smaller
than 5 nm, the uncertainties will be very large for $E$. We only calculated $E_{3\text{-}50}$ in this study and
$CoagSrc_{[i,j]}$ was thus neglected. Generally, $TR_{[i,j]}$ term cannot be quantified using the mathematic
method. As our previous study has indicated there was no significant transport term on a long-term
time scale (Kontkanen et al., 2020), we initially assumed that $TR_{[i,j]}$ equals 0. Another assumption is
that the influences from the variation in MLH are neglected. We briefly explore how the assumptions



of $TR_{[i,j]}$ and MLH influence the results in the next paragraph.
On the particle size distribution plots, we notice that the abrupt appearance of particles during non-
NPF days usually happens in the early morning (6:00-9:00) and late afternoon (17:00-20:00) (Figure
S1). The afternoon peak is accompanied by a decrease in MLH and an increase in particle numbers in
all sizes within the range of 3-50 nm, thus the abrupt increase in particle number concentration could
be due to the combined effects of MLH, transport, and emission. The morning peak is accompanied
by the increase in MLH, which should decrease particle number concentrations, and the increase in
particle number is only observed for 3-30 nm particles but not for 30-50 nm. Thus, the increasing
morning peak could only be caused by the primary emission of 3-30 nm particles. As a result, the 3-
30 nm particle emission rate during 6:00-9:00 is calculated to represent the average primary particle
emission rates for each day. It should be noted that the emission rates during 6:00-9:00 may be
underestimated due to the increase of MLH, and the emission rates only represent the increasing rates
of primary particles at the observational site, not the direct emission rates from the sources.
**3.  Results and discussion**
**3.1 UFP concentration, composition, and seasonal variability**
The overall concentration of UFPs is the highest in winter and the lowest in summer. The UFP number
concentrations   expressed   in   mean   ±   standard   deviation   are   $(1.7\pm1.2)\times10^4$,   $(1.5\pm1.1)\times10^4$,
$(1.1\pm0.7)\times10^4$, and $(1.5\pm0.9)\times10^4$ cm$^{-3}$ (Figure 1a) and the UFP mass concentrations are 1.3±0.9,
1.2±0.9, 1.0±0.6, and 1.2±0.7 µg m$^{-3}$ (Figure 1b) for winter, spring, summer, and autumn, respectively.
The seasonal variations are partly caused by the variation in MLH (Figure S2), while the other driving
factors are related to the source and loss terms of UFPs and will be further discussed in Section 3.3.

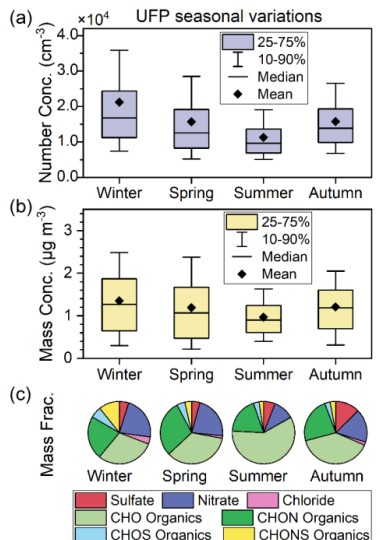




**Figure 1**. Seasonal variations of UFP concentrations and composition in urban Beijing. (a) UFP number concentrations in the size range of 3-100 nm. (b) UFP mass concentrations integrated from size distribution measurements, assuming spherical particles with a density of 1.4 g cm$^{-3}$. (c) Mass fractions of the components measured by the TDCIMS in negative ion mode.

The detected UFP composition is dominated by organics (68-81% for mass fraction), with minor contributions from nitrate (11-22%), sulfate (4-13%), and chloride (0.1-4%) over all four seasons (Figure 1c). The organic species include CHO, CHON, CHOS, and CHONS organics, contributing 30-59%, 19-29%, 3-6%, and 2-11% mass concentrations of the detected UFP compounds, respectively. The detected particulate species are similar for all four seasons as indicated by the similarities in the mass defect plots (Figure S3). The measured composition is consistent with the offline results from Beijing, which showed that organics were the most abundant in UFPs (Massling et al., 2009; Zhao et al., 2017). It should be noted that the collected mass integrated from the TDCIMS signals is ~50% of the total collected mass estimated from the size distributions (Figure S4). This is possibly due to the uncertainties in the quantification methods or because some UFP compounds (e.g., ammonia, amines, black carbon, and alkanes) cannot be ionized by $O_2^-$ in the TDCIMS. However, as the mass estimated from the two methods are in good correlation, we assume that the TDCIMS-measured composition is representative of UFP composition. As some of the particulate CHON, CHOS, and CHONS organics would decompose to CHO fragments in the TDCIMS during the thermal desorption process, there may be some underestimation of CHON and CHOS/CHONS organics and overestimation of CHO organics.

A major seasonal difference in UFP chemical composition is that the highest fractions of slow-desorbed CHO organics are observed in summer (59%), which may be related to the strongest solar radiation and lowest $NO_x$ concentrations. On the one hand, the appearance of most CHO organic ions during temperature ramping of the Pt wire during TDCIMS analysis occurs later, thus at a higher temperature, compared to nitrate and chloride, while slightly earlier, or at a lower desorption temperature, compared to sulfate (Figure S5). The higher temperature desorption, which we refer to as "slowly-desorbed," indicates that these species must be low-volatility compounds or the corresponding thermal decomposition fragments. On the other hand, the overall CHO organic mass has an afternoon peak at ~14:00, and its diurnal variation is consistent with that for $O_3$ in summer (Figure S6), indicating they might be related to photooxidation chemistry. Based on these, we hypothesize that CHO organics in the UFPs are mostly from the partitioning of low-volatility compounds originating from the gas-phase oxidation. Thus, the higher CHO fractions in summer are due to the strongest solar radiation, which benefits the gaseous photooxidation, and the lowest $NO_x$ (Figure S2), which contributes to the formation of CHO organics over CHON organics (Yan et al., 2020; Ye et al., 2019).

Another seasonal difference is that higher fractions of fast-desorbed species are measured in winter, including nitrate, chloride, and some CHON (e.g., $C_6H_4NO_3^-$, nitrophenols) organic compounds. These species are all desorbed at lower temperatures (Figure S5) and their concentrations in UFPs are



negatively correlated to ambient temperature (Figure S7), indicating their relatively higher volatility.
Thus, the higher fractions in winter are mainly governed by the temperature-dependent partitioning of
these compounds. It should be noted that CHONS organic (e.g., deprotonated aminomethanesulfonic
acid $CH_4NSO_3^-$ and deprotonated taurine $C_2H_6NSO_3^-$) fractions also increase in winter. Previously,
$CH_4NSO_3^-$ and $C_2H_6NSO_3^-$ were reported to be formed in the gas phase through the reaction between
$SO_3$ and amines under dry conditions (Li et al., 2018; Sarkar et al., 2019), which is favored in winter
Beijing where the ambient relative humidity is relatively low. The seasonal variations of these CHONS
species are different from those in larger particles where S-containing organics are mainly
organosulfates from primary emissions or heterogeneous/aqueous reactions (Ma et al., 2022).
The composition of UFPs also varies greatly with particle size. As shown in Figure 2, the most
significant size-dependent variations were observed for nitrate and CHO organics. The nitrate fraction
increases significantly with increasing particle diameter, probably due to the Kelvin effects that prevent
it from partitioning to small particles, or due to an increase in aqueous/heterogeneous processes at
larger particle sizes. The CHO organic fraction decreases significantly with increased particle size,
possibly due to its low volatility that favors smaller particles compared to the high-volatility
compounds. Compared to CHO organics, the relative contributions of N- and S-containing organics
increase with particle size, possibly due to higher volatility or the aqueous/heterogeneous formation as
particles grow. The sulfate fraction does not change significantly with particle sizes, possibly due to
the opposite size-dependence of condensational growth of $H_2SO_4$ and the aqueous/heterogeneous
formation of sulfate.

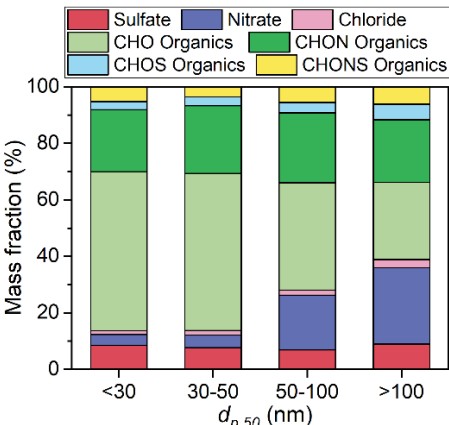


**Figure 2**. Size-dependent composition of UFPs. UFP composition mass fraction variation with the
representing particle size $d_{p,50}$. $d_{p,50}$ corresponds to 50% volume mean diameter of particles collected
on the TDCIMS filament.



**3.2 Sources of UFP organics and their seasonal variabilities**
As organics are the main components of UFPs, PMF source apportionments were performed for the
organic compounds. Five factors were identified in each season. The factor profiles and their diurnal
variations in winter are shown in Figure 3, and the results in other seasons are shown in Figure S8-10.
The correlations between PMF factors and key UFP components, trace gases, meteorology parameters,
and PM$_{2.5}$ for the four seasons are shown in Figure S11.

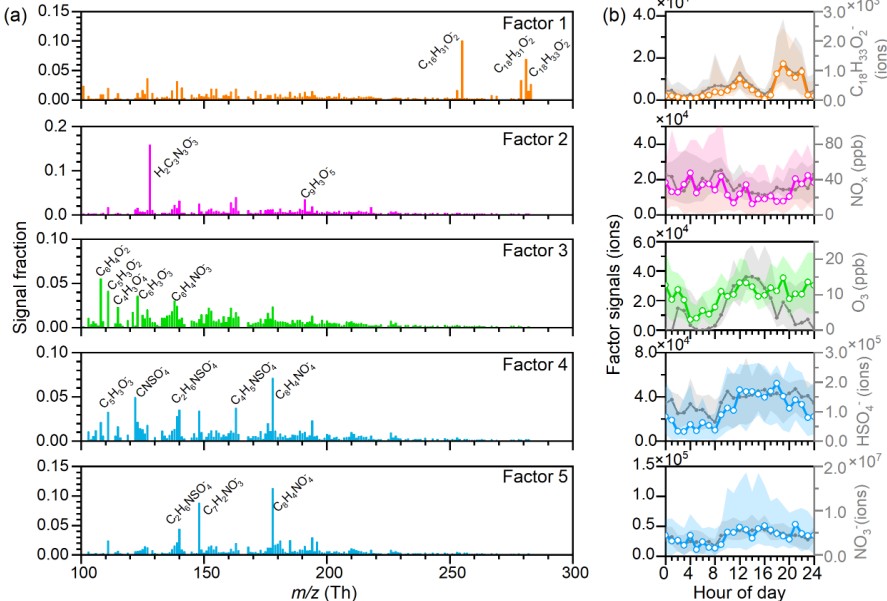


**Figure 3.** Source apportionment of the UFP organic composition (m/z 100-300) measured by the
TDCIMS in winter. (a) m/z profiles of the five PMF factors; (b) diurnal variations of each factor and
their related terms.
Factor 1 and factor 2 are identified as cooking-related and vehicle-related sources, respectively. Factor
1 is enriched in C$_{16}$H$_{31}$O$_2^-$, C$_{18}$H$_{31}$O$_2^-$, and C$_{18}$H$_{33}$O$_2^-$ (assigned to deprotonated palmitic acid, linoleic
acid, and oleic acids, respectively). Previous studies have revealed that saturated and unsaturated fatty
acids are the major constituents in cooking emissions, accounting for 73-85% of the cooking organic
matter, among which palmitic acid and oleic acid can be treated as the unique fingerprints of
atmospheric cooking particles (Zhao et al., 2007b, a). Factor 1 and the tracers show clear morning,
noon, and evening peaks, corresponding to breakfast, lunch, and dinner times. Factor 2 is enriched in
C$_3$N$_3$O$_3$H$_2^-$ (assigned to deprotonated cyanuric acid). Cyanuric acid was previously found with the
biggest emission in the urea-based selective catalytic reduction (SCR) technology for the reduction of
NO$_x$ from the exhaust of diesel-powered vehicles (Yassine et al., 2012). Factor 2 and the tracer show
clear morning peaks corresponding to the morning rush hours, consistent with the diurnal variation of



$NO_x$. These two factors are also identified in the other three seasons (Figure S8-10).
Besides these two primary sources, trace amounts of a biomass-burning tracer $C_6H_9O_5^-$ (assigned to
deprotonated levoglucosan) were also observed. However, its contribution to the total signal is small
and could not be separated into individual factors in the PMF analysis. We thus conclude that the
contribution of biomass burning to UFPs might be small in urban Beijing. This is understandable since
the burning of high-polluting fuels has been phased out in urban Beijing by the People's Government
of Beijing Municipality since 2014 (Municipality, 2014). Although large particles in urban Beijing
could be influenced by biomass burning and coal combustion through transport from surrounding
regions (Li et al., 2022b; Sun et al., 2015; Zhang et al., 2013; Zhang et al., 2008), UFPs could hardly
survive after long-distance transport due to their short lifetime.
Factors 3-5 are identified as secondary sources related to photooxidation formation or
aqueous/heterogeneous formation. In winter (Figure 3), factor 3 is enriched in slowly-desorbed, low-
volatility CHO organics and has daytime peaks at ~12:00-18:00, which is consistent with the diurnal
variation of $O_3$. They should come from gas-phase photooxidation followed by gas-particle
partitioning. Factor 4 and factor 5 are enriched in N- or S-containing organics. Their time series and
diurnal variability are highly correlated with sulfate, nitrate, $PM_{2.5}$, and relative humidity, indicating
the aqueous/heterogeneous formation pathway. Similarly, in spring and autumn, factor 3 is identified
as a photooxidation factor that is enriched in CHO organics, and factors 4-5 are identified as
aqueous/heterogeneous factors that are enriched in N- or S-containing organics (Figure S8-10).
Differently, in summer, factor 4 is identified as a photooxidation factor.
Clear seasonal variability of the sources was observed, with the contribution of primary emission
factors and aqueous/heterogeneous factors higher in winter and autumn, and the contribution of
photooxidation factors higher in summer (Figure S11). The sum of cooking and vehicle sources
contributed to 10-35% of the total organic signals in the m/z range of 100-300. The fractions of these
primary emissions are higher in winter and autumn, possibly indicating higher emissions. Another
possibility is that the oxidation degradation of these primary emissions is faster in summer and spring
due to higher oxidants and ambient temperature. The contributions of photooxidation factors are 20-
70% to the total organic signals in the m/z range of 100-300, with the highest in summer, and lowest
in winter. This is consistent with the highest CHO organic fractions in UFPs in summer in Figure 1
and is attributed to the strongest solar radiation. The contributions of aqueous/heterogeneous sources
are 15-60% to the total organic signals in the m/z range of 100-300, with the highest in winter and
lowest in summer.
To identify the sources for UFP numbers, we further combined the source analysis with variation in
particle size distributions. Among the identified four classes of composition sources, some are related
to the increase of UFP number concentrations, while others are related to the increase of UFP diameters.





The increase in UFP numbers is usually accompanied by the enhanced contribution of cooking- or
vehicle-related components or new particle formation events. An example is shown in Figure 4, a
relatively clean day with little interference from background aerosols. There are three periods where
UFP bursts were observed. During period 1, i.e., 6:00-9:00, a mode with a peak diameter at ~20 nm
appeared with a rapid increase in the vehicle tracer, $C_3N_3O_3H_2^-$. Compared to that before period 1, the
contribution of the vehicle-related factor increased from 7% to 25%. During period 2, i.e., 12:00-15:00,
new particle formation happens with a burst of particles at a peak diameter of 5-10 nm. Compared to
that before period 2, the contribution of the photooxidation-related factor increased from 64% to 92%.
This is consistent with our previous studies that slowly-desorbed CHO organics were the most
abundant compounds during NPF periods (Li et al., 2022a). During period 3, i.e., 18:00-22:00, a mode
with peak diameter at ~30 nm bursts, with a rapid increase in the cooking tracer, $C_{18}H_{31}O_2^-$. $C_{18}H_{31}O_2^-$
also has two minor peaks in the morning and noon time, consistent with the cooking activities.
Compared to that before period 3, the contribution of the cooking-related factor increased from 18%
to 67%. Thus, we conclude that the increase in UFP numbers in the three periods is mainly attributed
to the increase in vehicle emissions, new particle formations, and cooking emissions, respectively.

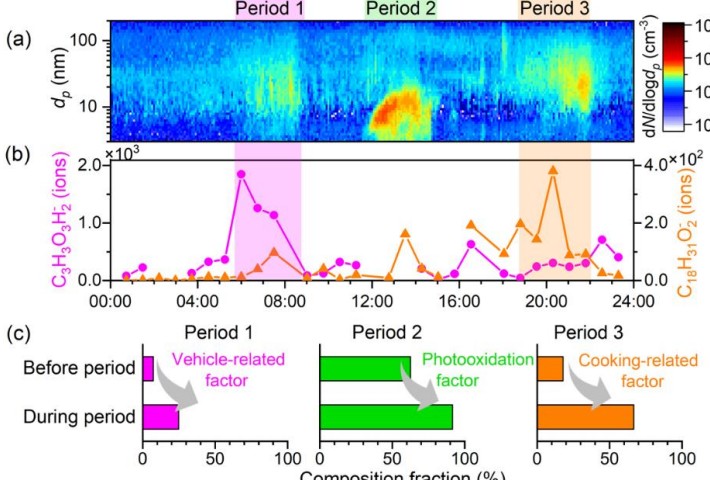


**Figure 4.** The particle number size distributions (a) and UFP composition variability (b-c) on April
16[th], 2020. The three periods with the abrupt appearance of UFP particles on this day are identified as
vehicle-related (period 1), NPF-related (period 2), and cooking-related (period 3) according to
TDCIMS composition measurements.
The morning, noon, and evening peaks in UFP numbers were widely observed during the observation
days in all four seasons. During non-NPF days, the UFP number concentration peak mainly appears in
the morning and evening time, corresponding to the primary emissions, and we choose the morning
periods to calculate the daily-averaged $E$. During NPF days, the UFP number peak mainly appears in



the noon time, and the daily-averaged $J$ was calculated during these periods. These further indicate that cooking emissions, vehicle emissions, and new particle formation are the main sources of UFP number concentrations.

Different from these three factors, the increased contribution of the aqueous/heterogeneous factor is not accompanied by the increase of UFP number concentrations but by the increase of UFP mode diameters. The contribution of aqueous/heterogeneous factor to sub-50 nm particles is only ~20%. For example, in the day presented in Figure 4, aqueous/heterogeneous factor accounted less than ~10% for the three bursts of UFP number concentration. However, it starts to dominate the organic composition when UFP particles grow above 50 nm (Figure S12), indicating an important role of aqueous/heterogeneous processes in the growth of particles larger than 50 nm in diameter.

**3.3 Driving factors for the seasonal variability of UFPs.**

As we have identified the main sources for UFP number concentrations in Section 3.2, we can further address the reasons for the significant seasonal differences in UFP number concentrations as has been shown in Figure 1, according to the variations in their sources and losses. The source terms mainly include new particle formation rates (here represented by $J_3$) and primary emission rates (here represented by $E_{3-50}$); the loss terms are presented by condensational growth rates ($GR_{3-50}$) and coagulation sinks. Here, we apply the condensation sink ($CS$) to evaluate the strength of coagulation loss.

The main sources of the UFP number concentration, $J_3$, and $E_{3-50}$, are both the highest in winter and the lowest in summer (Figure 5a-b), which are presumably caused by temperature effects. The temperature effect on $J_3$ is mainly due to the temperature-dependent cluster evaporation rates as reported in our previous study (Deng et al., 2020). This seasonal dependence of atmospheric UFPs attributed to vehicle emissions and its underlining reasons have not been revealed before. On the one hand, the low ambient temperature will largely increase the vehicle emission factors for particle numbers and gaseous hydrocarbons (Suarez-Bertoa and Astorga, 2018; Wen et al., 2021). On the other hand, a large fraction of the nanoparticles from vehicle emissions has been proposed to be formed by nucleation of the emitted hydrocarbon vapors or their oxidation products (Rönkkö and Timonen, 2019). The high ambient temperatures in summer may suppress the formation of these vehicle-related particles, just like it suppresses $J_3$ during ambient NPF.





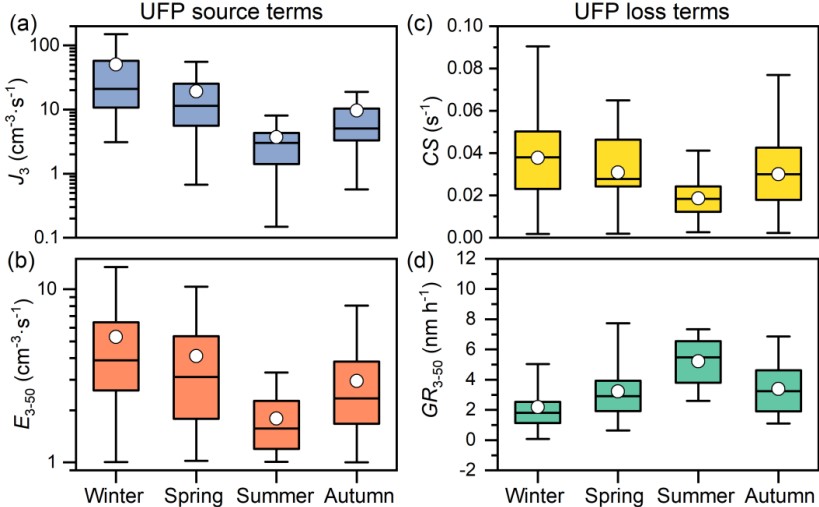

367

**Figure 5.** Seasonal variations of the main source and loss terms of UFP number concentration. (a) New
particle formation rates for 3 nm particles ($J_3$); (b) daily average primary particle emission rates for 3-
50 nm particles ($E_{3-50}$) during 6:00-9:00; (c) condensation sink ($CS$); (d) growth rates for 3-50 nm
particles ($GR_{3-50}$) during new particle formation events using the mode fitting method.

The lowest $GR$ of UFP occurs in winter (Figure 5d), which further contributes to high wintertime UFP
number concentrations. $CS$ and $GR_{3-50}$ have opposite trends, with $CS$ being the highest in winter while
$GR_{3-50}$ being the highest in summer (Figure 5c-d). The highest $GR_{3-50}$ in summer is due to the highest
condensable vapor concentrations in summer caused by strong solar radiation and high temperature
favoring the formation of condensable OOMs (Li et al., 2022a; Qiao et al., 2021). The theoretical
condensational $GR$ by OOMs and $H_2SO_4$ for 20 nm particles are 1.1, 3.0, 4.0, and 1.8 nm·h⁻¹ in winter,
spring, summer, and autumn, respectively, and they are close to $GR$ derived during NPF events using
the mode-fitting method as shown in Figure 5d. Under the estimated $GR$, the time needed for sub-3
nm to grow above 50 and 100 nm is the shortest in summer (9 and 19 h, respectively), and the longest
in winter (24 and 49 h, respectively). Thus, the lower GR in winter also contributes to the highest UFP
number concentrations in winter.

## 4. Conclusions

In this study, we explored the UFP composition and sources in typical polluted urban environments
based on near-continuous measurements of UFP composition and size distributions in Beijing over
four seasons. We observed that UFP composition varied with seasons and particle diameter, indicating
their different sources. Specifically, photooxidation processes generate more CHO organics, leading
to higher CHO fractions in summer. While aqueous/heterogeneous processes generate more N- and S-
containing organics, leading to higher N- and S-containing organic fractions in above-50 nm particles



than sub-50 nm particles. Combining the PMF analysis for UFP organics and the size distribution
analysis, we found that vehicle and cooking emissions are two of the most important primary sources
of UFP number concentrations in urban Beijing, while new particle formation is the most important
secondary source of UFP number concentrations and would increase the contribution of CHO organics
to UFP composition. The aqueous/heterogeneous sources would not increase UFP number
concentration but would increase UFP mode diameters and mass concentrations. For the seasonal
variations, we found that UFP number concentrations are the highest in winter. This is mainly due to
the highest primary particle emissions, the highest new particle formation rates, and the lowest particle
growth rates in winter. Further controlling of UFPs in urban Beijing needs to focus on vehicle
emissions, and the gas precursors related to secondary sources of UFPs.
The observed distinct seasonal variabilities of UFP composition and their size dependence emphasize
the importance of long-term and high-time-resolution measurements of both UFP composition and size
distributions. This could provide valuable datasets for the evaluation of UFP's long-term exposure risks.
The high time-resolution measurements combined with PMF analysis can also help identify the
secondary UFP sources, which contribute the major fraction of organic signals but could not be
identified from previous offline UFP measurements. Further addressing the UFP composition and
sources on the regional scale still requires measurements at sites with different distances from the
emission sources due to the short lifetime of UFPs.

**Data availability.**
Data are available upon request from the corresponding authors.
**Supplement.**
The contents of the supporting information include the diurnal variations of $E_{3\text{-}50}$ during non-NPF days
over four seasons (Figure S1); the diurnal variations of MLH, UVB, T, RH, $O_3$, $NO_x$, and $PM_{2.5}$ in the
four seasons (Figure S2); details of the measured UFP composition during four seasons (Figure S3);
seasonal variation of the UFP mass estimated from TDCIMS and PSD (Figure S4); the averaged
thermal desorption profiles of different UFP composition (Figure S5); the diurnal variation of CHO
organics in the four seasons (Figure S6); temperature dependence of some fast-desorbed UFP
composition (Figure S7); the spectra of five PMF-factors during spring, summer, and autumn (Figure
S8-10); summary of PMF factors during the four seasons (Figure S11); the contribution of different
factors as a function of particle sizes (Figure S12); summary of sampling periods (Table S1).
**Author contributions.**
XL, JJ, and JS designed the study. XL, YC, YYL, RC, YRL, and CD participated in data collection



and performed the data analysis. XL prepared the manuscript with contributions from all co-authors.
All authors approved the final version of the manuscript.
**Competing interests.**
The authors declare that they have no conflict of interest.
**Financial supports.**
Financial support from the National Natural Science Foundation of China (22188102 and 22106083),
Samsung PM$_{2.5}$ SRP is acknowledged. JS acknowledges funding from the US Department of Energy
(DE-SC0021208) and the US National Science Foundation (CHE-2004066).

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
