# Peer review of "Seasonal variations in composition and sources of atmospheric ultrafine particles"

_EGUsphere, 2023_

## Referee Comment (RC2)

Referee comment on "Seasonal variations in composition and sources of atmospheric ultrafine particles in urban Beijing based on near-continuous measurements" by Li et al., https://doi.org/10.5194/egusphere-2023-809

Li et al., presented an extensive study on the chemical composition of ultra fine particles in an urban area in Beijing. The measurements were performed over the four seasons, which makes this study very robust. The chemical composition of particles was measured by a thermal desorption chemical ionization mass spectrometer (TDCIMS). The authors found that the particles measured in Beijing are dominated by organic compounds. CHO organic compounds are the main constituents of particles during summer, while organic particles containing sulfur, nitrogen, nitrate, and chloride are more abundant in winter. A Positive Matrix Factorization (PMF) analysis was performed in order to determine the sources of these particles. A 5-Factor solution was suggested, and the authors related these factors to cooking, vehicle emission, photooxidation formation, and aqueous/heterogeneous sources.

Besides the chemical composition, the authors reported nucleation, growth, and emission rates. Thus, the highest particle number concentrations were found in winter. The authors explain this fact due to the highest primary particle emission, the highest new particle formation, and the lowest growth rates in winter.

The study by Li et al., is highly valuable and certainly contributes to the understanding of urban ultrafine particles. I appreciate the way in which the manuscript is written. I very much enjoyed the reading. Furthermore, I would recommend it to be published on EGUsphere after addressing the following comments.

Specific comments:

Lines 17, 19, and along the manuscript: I would recommend using either UFPs or UFP to make the manuscript uniform.

Could the authors define CHO in the abstract? once in the abstract and once in the introduction.

Why the collection efficiency on the filament decreases with the particle size?

Is the $O_2^-$ chemical ionization technique more sensitive towards low and/or high oxygenated organic compounds?

At which temperature is the filament heated up? Is this done gradually, how long does it take to evaporate the sample? Is the filament heated directly, or is there any heated carrier flow? Is the highest temperature enough for desorbing all the compounds collected?

How do the authors determine that a large fraction of the compounds below 100 m/z were produced by thermal decomposition?

In line 141: is this the normalized signal (by the reagent ions) or is the raw signal?

In Eq. (1) is the GR the net condensation growth term or is the condensation growth rate of the particle?

In line 106 is written that the TDCIMS performed the measurements in the "bulk collection mode", with this I assume the TDCIMS does not select previously the size before collection and evaporation. In this sense, can the authors clarify how the size-dependent composition then is done? Is this analysis base on the observed size distribution measured by the PSD and SMPS? Or the PSD and SMPS were coupled to the TDCIMS allowing the collection of particles with known size?

If the PSD and/or SMPS were not coupled to the TDCIMS. Did the authors observe size distributions with 2 or 3 modes? How did the authors isolate the effect of the big particle on the small particles?

What are the possible losses that the TDCIMS can experience?

In lines 218 and 219, what does slow desorbed compounds mean, in terms of temperature? And why their appearance is related to the highest solar radiation and low NOx? Could the authors clarify how slow and fast desorbed compounds are defined in terms of temperature before explaining further their characteristics?

In line 220: what does "occurs later" mean here?

Lines 237-240: can the authors add connectors or make these sentences shorter?

Page 10: I very much appreciate the interpretation of the PMF analysis. I have a couple of questions about this. How do the residuals look like? Do the authors observe any factor related to background, or was the background removed from the data?

Did the authors apply any complementary technique for fully identifying the compounds described in Section 3.2? Is it possible that any of these compounds, for example, $C_3N_3O_3H_2^-$, and $C_6H_9O_5^-$ are affected to some extent by thermal decomposition and not fully correspond to cyanuric acid and deprotonated levoglucosan? Do the authors characterize thermally these compounds, so they can certainly claim that these are cyanuric acid and deprotonated levoglucosan?

---

## Author Comment (AC1)

**Responses to Reviewers' Comments on Manuscript egusphere-2023-809**
(Seasonal variations in composition and sources of atmospheric ultrafine particles in urban Beijing based on near-continuous measurements)

We are grateful for the reviewers' comments and we feel that our responses to these have improved this manuscript. We have addressed the comments in the following paragraphs and made corresponding changes in the revised manuscript. Comments are shown as *blue italic text* followed by our responses. Changes are highlighted in the revised manuscript and shown as "quoted underlined text" in our responses.

**Reviewer #1:**
*This is a well written paper on the sources and growth of ultrafine particles in urban air. Chemical composition measurements assist the identification of UFP sources, which include both primary emissions and new particle formation. Aqueous/heterogeneous growth of preexisting particles is also indicated from chemical composition data. Measured particle growth rates (3-50 nm) during new particle formation are consistent with theoretical growth rates estimated from condensation of gas-phase sulfuric acid and low-volatility organic compounds.*

*Specific comments:*
*In the next to the last sentence of Section 3.3, the authors estimate the times needed for sub-3 nm particles to grow to above 50 and 100 nm. The authors do not explicitly state how these times are calculated, but I am assuming that they are based on the measured and/or estimated 3-50 nm growth rates. If this is the case, then the times they give to grow above 50 nm are accurate, but the times they give to grow above 100 nm are likely to be a substantial overestimate. Figure S12 shows that the CHON/S composition factor associated with aqueous/heterogeneous chemistry becomes very large above 50 nm, roughly equaling that of the CHO-rich factor which presumably includes (butis not limited to) condensation of low volatility organics. Therefore, it is likely that particle growth due to aqueous/heterogeneous chemistry equals or exceeds particle growth due to condensation alone. If both growth channels are taken into account, the actual time to grow to 100 nm is likely to be much shorter than what the authors report. I encourage the authors to consider this possibility (or to clarify how the times were calculated) and discuss appropriately in the body of the paper.*

**Response:** We agree with the reviewer that the time estimated for $<3$ nm particles to grow above 100 nm was possibly overestimated. The time was estimated from the measured average $GR_{3\text{-}50}$, the latter of which is derived from the observed particle size distribution, and we used $GR_{3\text{-}50}$ for the estimation of the size range between 3 and 100 nm. This would probably overestimate the time needed to grow above 100 nm, as our data have suggested growing $GR$ with increasing $d_p$. The increasing contribution from aqueous/heterogeneous chemistry above 50 nm also suggests that $GR_{50\text{-}100}$ are larger than $GR_{3\text{-}50}$. However, it is difficult to determine $GR_{50\text{-}100}$ from the measured size distribution for most of the observed NPF events. The main reason is that the new particle formation events was usually disturbed during ~16:00-20:00 by primary particle emission (cooking and vehicle emission) and the decrease of boundary layer. Before 16:00-20:00, the particle mode diameters usually cannot grow above ~50 nm. Furthermore, when particles grow to ~70-80 nm, they start to mix with the background particles. These influence associated with an inhomogeneous environment made it difficult to retrieve $GR_{50\text{-}100}$ from the measured particle size distributions.

In the revised manuscript, we removed the time needed to grow above 100 nm (as it is hard to estimate). The sentence is revised as:

[Line 383-385] Under the observed average GR$_{3-50}$, the time needed for sub-3 nm to grow above 50 nm is the shortest in summer (~9 h, respectively) and the longest in winter (~24 h).

*A related question: What is the approximate survival probability of a sub-3 nm particle growing to 50 or 100 nm without being removed by a typical loss process? It seems the authors dataset is robust enough to provide an estimate.*

**Response:** We thank the reviewer for this inspiring comment. Based on case-by-case analysis of previous dataset from the same measurement site, we estimated that the median survival probability of a 3 nm particle growing to 100 nm was ~1 %, with good consistency between the measurement and theory of growth and coagulation scavenging (Cai et al., 2022; Li et al., 2022). For example, the survival probability of the 8 nm particles to 50 nm is ~0.8% during the new particle formation and growth events in Oct. 2$^{nd}$, 2020. However, not every NPF event has such a clear growth shape. The calculation of survival probability of new particles in each NPF events needs to carelly isolate the newly formed particles from the background particles, and batch calculations require plenty of surpervised work. The seansonal variations of CS and GR shown in Fig. 5 in this manuscript indicate that the survival probability may vary with seasons. Combining careful analysis of every NPF events and our robust dataset, we think it will be an interesting future work to provide an estimate of survival probability under different environmental conditions (e.g. seasons).

[Figure]

Figure R1. The growth and survival of new particles in Beijing, October 2, 2020. (a) Size distributions of particles and the mode diameters of new particles from October 2, 2020, to October 3, 2020. (b) The survival ratios of new particles. The measured survival ratio of the 8 nm particles to 50 nm was ~0.8%, close to the theoretical survival ratio (~0.5%) due to coagulation loss. (Li et al., 2022)

References

Li, X., Li, Y., Cai, R., Yan, C., Qiao, X., Guo, Y., Deng, C., Yin, R., Chen, Y., Li, Y., Yao, L., Sarnela, N., Zhang, Y., Petäjä, T., Bianchi, F., Liu, Y., Kulmala, M., Hao, J., Smith, J. N., and Jiang, J.: Insufficient Condensable Organic Vapors Lead to Slow Growth of New Particles in an Urban Environment, Environ. Sci. Technol., 55, 9936-9946, 10.1021/acs.est.2c01566, 2022.

Cai, R., Deng, C., Stolzenburg, D., Li, C., Guo, J., Kerminen, V., Jiang, J., Kulmala, M., Kangasluoma, J.: Survival probability of new atmospheric particles: closure between theory and measurements from 1.4 to 100 nm, Atmos. Chem. Phys., 22, 14571-14587, 10.5194/acp-22-14571-2022, 2022.

---

## Author Comment (AC2)

**Responses to Reviewers' Comments on Manuscript egusphere-2023-809**
(Seasonal variations in composition and sources of atmospheric ultrafine particles in urban Beijing based on near-continuous measurements)

We are grateful for the reviewers' comments and we feel that our responses to these have improved this manuscript. We have addressed the comments in the following paragraphs and made corresponding changes in the revised manuscript. Comments are shown as *blue italic text* followed by our responses. Changes are ==highlighted== in the revised manuscript and shown as "quoted underlined text" in our responses.

**Reviewer #2:**

*Li et al., presented an extensive study on the chemical composition of ultrafine particles in an urban area in Beijing. The measurements were performed over the four seasons, which makes this study very robust. The chemical composition of particles was measured by a thermal desorption chemical ionization mass spectrometer (TDCIMS). The authors found that the particles measured in Beijing are dominated by organic compounds. CHO organic compounds are the main constituents of particles during summer, while organic particles containing sulfur, nitrogen, nitrate, and chloride are more abundant in winter. A Positive Matrix Factorization (PMF) analysis was performed in order to determine the sources of these particles. A 5-Factor solution was suggested, and the authors related these factors to cooking, vehicle emission, photooxidation formation, and aqueous/heterogeneous sources.*

*Besides the chemical composition, the authors reported nucleation, growth, and emission rates. Thus, the highest particle number concentrations were found in winter. The authors explain this fact due to the highest primary particle emission, the highest new particle formation, and the lowest growth rates in winter.*

*The study by Li et al., is highly valuable and certainly contributes to the understanding of urban ultrafine particles. I appreciate the way in which the manuscript is written. I very much enjoyed the reading. Furthermore, I would recommend it to be published on EGUsphere after addressing the following comments.*

*Specific comments:*
1. *Lines 17, 19, and along the manuscript: I would recommend using either UFPs or UFP to make the manuscript uniform.*

**Response:** Thanks for the suggestion. We checked through the manuscript and made sure that "UFPs" was used when it is a noun, while "UFP" was used when it is used in a noun phrase, such as "UFP concentration", "UFP composition".

2. *Could the authors define CHO in the abstract? once in the abstract and once in the introduction.*

**Response:** Corrected as "CHO organics (i.e., molecules containing carbon, hydrogen, and oxygen)".

3. *Why the collection efficiency on the filament decreases with the particle size?*

**Response:** The particle collection on the high voltage-biased Pt filament is based on electrostatic precipitation. As smaller particles have larger electrical mobility ($Z_p \propto 1/d_p$), they are easier to go across the $N_2$ protection sheath and to be collected on the filament. The measured electrostatic

collection efficiency for atomized (NH₄)₂SO₄ particles and for ambient particles are shown in Figure R1.

The electrostatic collection scenario is similar to that of a tube-wire electrostatic precipitator where collection efficiency decreases with growing particle size. The Deustch-Anderson equation(Deutsch, 1922) is often used to evaluate the collection efficiency of a tube-wire electrostatic precipitator:

$$\eta = 1 - e^{-\frac{c_{ES}A_c}{Q}} \tag{R1}$$

where $\eta$ is the collection efficiency; $c_{ES}$ is the migrating velocity near the wall of the tube, in proportion to $d_p^{-1.5}$; $A_c$ is the collection area of the wall and $Q$ is the flow rate. Thus the collection efficiency depends on particle diameter ($d_p$) as $1 - e^{-kd_p^{-1.5}}$. Therefore, equation R2 was used to fit the experimental data in Figure R2,

$$CE = 1 - e^{(-k \cdot d_p^{-b})} \tag{R2}$$

For the atomized (NH₄)₂SO₄ particles, the fitted parameters $k$ and $b$ are 734.4 and 2.8, respectively. A good correlation was achieved with an R-value of 0.996 (Figure R1).

[Figure]

Figure R1. The collection efficiency for atomized (NH₄)₂SO₄ particles and ambient particles. Dots are the measured values and lines are the fitted curves. (Li et al., 2021)

We added the explanation in the manuscript:

[Line 110-112] The particle electrostatic collection efficiency on the filament decreases rapidly with increasing particle size due to decreased electrical mobility, ensuring that the collected particle mass is mainly from UFPs (Li et al., 2021).

4. *Is the O2 - chemical ionization technique more sensitive towards low and/or high oxygenated organic compounds?*

**Response:** Most organic compounds measured with O₂⁻ are with 2-5 oxygen atoms. So, O₂⁻ is more sensitive towards high oxygenated organics compared to H₃O⁺, while is less sensitive towards high oxygenated organics compared to NO₃⁻. The sentence are revised as:

[Line 115-118] Using O₂⁻ as the reagent ion, sulfate, nitrate, chloride, and most of the oxygenated organics can be measured, while black carbon, hydrocarbon compounds, and bases such as ammonium and aminium cannot are less likely to be detected due to lower sensitivity.

5. *At which temperature is the filament heated up? Is this done gradually, how long does it take to evaporate the sample? Is the filament heated directly, or is there any heated carrier flow? Is the highest temperature enough for desorbing all the compounds collected?*

**Response:** The details of the TDCIMS operation are given in our previous studies. The filament is

heated directly through applying a current to it. The temperature rises to the maximum within 35 s and is maintained for 15 s. Most compounds, including sulfate, can desorb thoroughly within this period. For clarification, the following sentences are added to the manuscript:

[Line 113-115] During the particle evaporation, an electrical current is applied to the metal filament to an estimated temperature of ~600 °C within a minute. The observed compounds can be desorbed thoroughly within the heating periods as indicated by the desorption profile.

[Figure]

Figure R2. The thermal desorption profiles for some compounds detected in the positive ion mode or the negative ion mode in urban Beijing. Solid lines are the sample signals and dashed lines are the corresponding background signals. Gray lines are the electrical current applied to the Pt filament.

6. *How do the authors determine that a large fraction of the compounds below 100 m/z were produced by thermal decomposition?*

**Response:** Firstly, we have calibrated ~30 organic compounds and found that some oxygenated compounds would decompose to smaller fragments during the heating, especially for the highly oxygenated organics. Secondly, we estimated the volatility of the observed below-100 m/z compounds through the molecular formula, and found that the volatility of those are not enough to stay in particle phase by themselves, unless they are fragments from lower-volatility compounds. Though a fraction of these compounds could be organic acids and stay in the particle phase through salt formation, most of them are not acids. Thirdly, the desorption time of most organics below m/z 100 are slow, indicating they might be decomposition products. Thus, we assume that most of these compounds are fragmentations.

7. *In line 141: is this the normalized signal (by the reagent ions) or is the raw signal?*

**Response:** The signals are normalized by the variation of reagent ions to take into account the instrument variation. The details of the data analysis are given in our previous study (Li et al., 2021).

8. *In Eq. (1) is the GR the net condensation growth term or is the condensation growth rate of the particle?*

**Response:** GR (m s$^{-1}$)is the condensational growth rate of particles, and GR($n_j-n_i$) (m$^{-3}$ s$^{-1}$) is the net condensation growth term.

9. *In line 106 is written that the TDCIMS performed the measurements in the "bulk collection*

*mode", with this I assume the TDCIMS does not select previously the size before collection and evaporation. In this sense, can the authors clarify how the size-dependent composition then is done? Is this analysis base on the observed size distribution measured by the PSD and SMPS? Or the PSD and SMPS were coupled to the TDCIMS allowing the collection of particles with known size?*

**Response:** This question is related to Question 3, the electrostatic collection efficiency of the metal filament decrease with particle size (as shown in Figure R1 and Figure R2). For example, in urban Beijing, when the Nano DMA is turned off, more than 70% of the collected mass is from sub-100 nm (Figure R3). The "size-dependent composition" analysis in Figure 2 is based on the representative diameter $d_{p,50}$ (50% volume mean diameter of particles collected on the TDCIMS filament) instead of the selected diameter. The estimation is based on an independent PSD multiplying the sampling efficiencies of the particles with different sizes (Figure R2).

[Figure]

Figure 2. The TDCIMS sampling efficiencies as a function of the particle size when the upstream Nano DMAs are turned off. The PE in the sampling tubes was theoretically calculated. The UPCE was measured previously(Chen et al., 2019) (blue points) and fitted (blue dashed line). The CE was measured (green points) and fitted (green dashed line). The overall particle sampling efficiency (black dashed line) was calculated by multiplying the PE, UPCE, and CE.

[Figure]

Figure R3. (a) The average aerosol size distribution of non-NPF days during the whole sampling period. (b) The average number distribution of particles estimated to be collected on the TDCIMS filament before and after considering multiple charges. (c) The average mass distribution of particles (assuming spherical particles with a density of 1.4 g·cm-3) estimated to be collected on the TDCIMS

filament before and after considering multiple charges. (Li et al., 2021)

10. *If the PSD and/or SMPS were not coupled to the TDCIMS. Did the authors observe size distributions with 2 or 3 modes? How did the authors isolate the effect of the big particle on the small particles?*

**Response:** Yes, sometimes there are two modes on the size distribution plot. But usually the second mode is larger than 100 nm. As shown in Figure R3, the particles collected on the filament are mostly ultrafine particles. The influences from >100 nm particles are minor.

11. *What are the possible losses that the TDCIMS can experience?*

**Response:** As shown in Figure R2, the particle charging efficiency, the collection efficiency, and the penetration efficiency in the sampling tubes determine the overall particle sampling efficiency.

12. *In lines 218 and 219, what does slow desorbed compounds mean, in terms of temperature? And why their appearance is related to the highest solar radiation and low NOx? Could the authors clarify how slow and fast desorbed compounds are defined in terms of temperature before explaining further their characteristics?*

**Response:** The slow-desorbed compounds indicate the desorption peak is slower than nitrate. They may corresponds to the condensation of highly oxygenated organic gas molecules (HOMs). It is reported previously that the formation of gaseous HOMs is promoted by increased solar radiation while is suppressed by increased $NO_x$. We clarified the explanation of slow-desorbed compounds in the manuscript:

[Line 223-226] On the one hand, the appearance of most CHO organic ions during temperature ramping of the Pt wire occurs at higher temperatures compared to nitrate and chloride, while at slightly lower temperatures compared to sulfate (Figure S5)

[Figure]

Figure S5. Averaged, normalized thermal desorption profiles of (a) the slowly desorbed compounds and (b) the quickly desorbed compounds. The signals are normalized to the corresponding highest signal of the thermal desorption curves.

13. *In line 220: what does "occurs later" mean here?*

**Response:** "occurs later" means "desorb latter" during the temperature ramping process, as shown in Figure S5. We corrected it as "occurs at higher temperatures".

*14. Lines 237-240: can the authors add connectors or make these sentences shorter?*

**Response:** Corrected.

[Line 241-244] Previously, $CH_4NSO_3^-$ and $C_2H_6NSO_3^-$ were reported to be formed in the gas phase through the reaction between $SO_3$ and amines under dry conditions (Li et al., 2018; Sarkar et al., 2019). Their gaseous formation likely happens in winter Beijing due to low ambient relative humidity.

*15. Page 10: I very much appreciate the interpretation of the PMF analysis. I have a couple of questions about this. How do the residuals look like? Do the authors observe any factor related to background, or was the background removed from the data?*

**Response:** The residues are smaller than 8%. Firstly, the background particles larger than 150 nm are not likely to be detected due to the decreasing particle collection efficiency on the Pt filament as mentioned in Questions 3 and 9. So the influences from the background particles might be small. Secondly, the aqueous/heterogeneous factors are highly related to the background aerosol from its high correlation with $PM_{2.5}$, a fraction of these factors might be from the background aerosol, while another fraction is from the aqueous/heterogeneous reactions in UFPs.

*16. Did the authors apply any complementary technique for fully identifying the compounds described in Section 3.2? Is it possible that any of these compounds, for example, C3N3O3H2 - , and C6H9O5 - are affected to some extent by thermal decomposition and not fully correspond to cyanuric acid and deprotonated levoglucosan? Do the authors characterize thermally these compounds, so they can certainly claim that these are cyanuric acid and deprotonated levoglucosan*

**Response:** Thanks for the suggestion. We have not done complementary techniques for identifying these compounds. For $C_3N_3O_3H_2^-$, it's not likely fragment because few larger ion has similar timeserie with $C_3N_3O_3H_2^-$. For $C_6H_9O_5^-$, there may be contributions from decomposition. However, as biomass burning is not identified as a major source of UFPs in Beijing, we do not rely on this marker to draw any conclusion. We agree that characterizations of the standard compounds in the TDCIMS will help improve our understanding, and we will do the calibrations in future studies.

References

Chen, X., Jiang, J., and Chen, D.-R.: A soft X-ray unipolar charger for ultrafine particles, J. Aerosol. Sci., 133, 66-71, 10.1016/j.jaerosci.2019.04.010, 2019.

Deutsch, W.: Bewegung und Ladung der Elektrizitätsträger im Zylinderkondensator, Annalen der Physik, 373, 335-344, 10.1002/andp.19223731203, 1922.

Li, X., Li, Y., Lawler, M. J., Hao, J., Smith, J. N., and Jiang, J.: Composition of Ultrafine Particles in Urban Beijing: Measurement Using a Thermal Desorption Chemical Ionization Mass Spectrometer, Environ. Sci. Technol., 55, 2859–2868, 10.1021/acs.est.0c06053, 2021.

Li, X., Li, Y., Cai, R., Yan, C., Qiao, X., Guo, Y., Deng, C., Yin, R., Chen, Y., Li, Y., Yao, L., Sarnela, N., Zhang, Y., Petäjä, T., Bianchi, F., Liu, Y., Kulmala, M., Hao, J., Smith, J. N., and Jiang, J.: Insufficient Condensable Organic Vapors Lead to Slow Growth of New Particles in an Urban Environment, Environ. Sci. Technol., 55, 9936-9946, 10.1021/acs.est.2c01566, 2022.